# Physical activity and risks of breast and colorectal cancer: a Mendelian randomisation analysis

Nikos Papadimitriou et al.[#]

Physical activity has been associated with lower risks of breast and colorectal cancer in epidemiological studies; however, it is unknown if these associations are causal or confounded. In two-sample Mendelian randomisation analyses, using summary genetic data from the UK Biobank and GWA consortia, we found that a one standard deviation increment in average acceleration was associated with lower risks of breast cancer (odds ratio [OR]: 0.51, 95% confidence interval [CI]: 0.27 to 0.98, P-value = 0.04) and colorectal cancer (OR: 0.66, 95% CI: 0.48 to 0.90, P-value = 0.01). We found similar magnitude inverse associations for estrogen positive (ER$^{+ve}$) breast cancer and for colon cancer. Our results support a potentially causal relationship between higher physical activity levels and lower risks of breast cancer and colorectal cancer. Based on these data, the promotion of physical activity is probably an effective strategy in the primary prevention of these commonly diagnosed cancers.

---

[#]A full list of authors and their affiliations appears at the end of the paper.

Breast and colorectal cancer are two of the most common cancers globally with a combined estimated number of 4 million new cases and 1.5 million deaths in 2018[1]. Physical activity is widely promoted along with good nutrition, maintaining a healthy weight, and refraining from smoking, as key components of a healthy lifestyle that contribute to lower risks of several non-communicable diseases such as cardiovascular disease, diabetes, and cancer[2].

Epidemiological studies have consistently observed inverse relationships between physical activity and risks of breast and colorectal cancer[3–5]. The World Cancer Research Fund/American Institute for Cancer Research (WCRF/AICR) Continuous Update Project classified the evidence linking physical activity to lower risks of breast (postmenopausal) and colorectal cancer as 'strong'[6]. However, previous epidemiological studies have generally relied on self-report measures of physical activity which are prone to recall and response biases and may attenuate 'true' associations with disease risk[7]. More objective methods to measure physical activity, such as accelerometry, have seldom been used in large-scale epidemiological studies, with the UK Biobank being a recent exception in which ~100,000 participants wore a wrist accelerometer for 7-days to measure total activity levels[8]. Epidemiological analyses of these data will provide important new evidence on the link between physical activity and cancer, but these analyses remain vulnerable to other biases of observational epidemiology such as residual confounding (e.g. low physical activity levels may be correlated with other unfavourable health behaviours) and reverse causality (e.g. preclinical cancer symptoms may have resulted in low physical activity levels).

Mendelian randomisation (MR) is an increasingly used tool that uses germline genetic variants as proxies (or instrumental variables) for exposures of interest to enable causal inferences to be made between a potentially modifiable exposure and an outcome[9]. Unlike traditional observational epidemiology, MR analyses should be largely free of conventional confounding owing to the random independent assignment of alleles during meiosis[10]. In addition, there should be no reverse causation, as germline genetic variants are fixed at conception and are consequently unaffected by the disease process[10].

We used a two-sample MR framework to examine potential causal associations between objective accelerometer-measured physical activity and risks of breast and colorectal cancer using genetic variants associated with accelerometer-measured physical activity identified from two recent genome-wide association studies (GWAS)[11,12]. We examined the associations of these genetic variants with risks of breast cancer[13] and colorectal cancer[14].

## Results

**MR estimates for breast cancer.** We estimated that a 1 standard deviation (SD) (8.14 milligravities) increment in the genetically predicted levels of accelerometer-measured physical activity was associated with a 49% lower risk of breast cancer for the instrument using the 5 genome-wide-significant SNP instrument (odds ratio [OR]: 0.51, 95% confidence interval [CI]: 0.27 to 0.98, P-value = 0.04, Q-value = 0.062) (Table 1), and a 41% lower risk for the extended 10 SNP instrument (OR: 0.59, 95% CI: 0.42 to 0.84, P-value = 0.003, Q-value = 0.012). An inverse association was only found for estrogen receptor positive breast cancer (ER[+ve]) (5 SNP instrument, OR: 0.45, 95% CI: 0.20 to 1.01, P-value = 0.054, Q-value = 0.077; extended 10 SNP instrument, OR: 0.53, 95% CI: 0.35 to 0.82, P-value = 0.004, Q-value = 0.004), and not estrogen receptor negative (ER[-ve]) breast cancer (Table 1); although this heterogeneity by subtype was not statistically different ($I^2 = 16\%$; P-heterogeneity by subtype = 0.27). There was some evidence of heterogeneity based on Cochran's Q

(P-value < 0.05) for the breast cancer analyses; consequently, for these models random effects MR estimates were used (Table 1). MR estimates for each of the SNPs associated with accelerometer-measured physical activity in relation to breast cancer risk are presented in Fig. 1 and Supplementary Fig. 1. Scatter plots (with coloured lines representing the slopes of the different regression analyses) and funnel plots of the accelerometer-measured physical activity and breast cancer risk association for the extended 10 SNP instrument are presented in Supplementary Figs. 2 and 3.

**Mendelian randomisation estimates for colorectal cancer.** For colorectal cancer, a 1 SD increment in accelerometer-measured physical activity level was associated with a 34% lower risk (OR: 0.66, 95% CI: 0.48 to 0.90, P-value = 0.01, Q-value = 0.022) for the 5 SNP instrument, and a 40% lower risk for the extended 10 SNP instrument (OR: 0.60, 95% CI: 0.47 to 0.76, P-value = $2.4 \times 10^{-5}$, Q-value = 0.0002) (Table 1). The inverse effect estimate was stronger for women (OR: 0.57, 95% CI: 0.36 to 0.90, P-value = 0.02, Q-value = 0.036), while there was weak evidence for an inverse association for men (OR: 0.79, 95% CI: 0.50 to 1.23, P-value = 0.29, Q-value = 0.31); this heterogeneity did not meet the threshold of significance ($I^2 = 0\%$; P-heterogeneity by sex = 0.34). For colorectal subsite analyses, accelerometer-measured physical activity levels were inversely associated with risks of colon cancer (OR per 1 SD increment OR: 0.64, 95% CI: 0.44 to 0.94, P-value = 0.02, Q-value = 0.036); while there was weak evidence for an inverse association between accelerometer-measured physical activity levels and rectal cancer (OR: 0.70, 95% CI: 0.43 to 1.14, P-value = 0.15, Q-value = 0.18). Similar results by sex and subsite for colorectal cancer were found for the extended 10 SNP instrument (Table 1). MR estimates for each individual SNP associated with accelerometer-measured physical activity in relation to colorectal cancer risk are presented in Fig. 2 and Supplementary Figs. 4–6. Scatter plots (with coloured lines representing the slopes of the different regression analyses) and funnel plots of the accelerometer-measured physical activity and colorectal cancer risk association for the extended 10 SNP instrument are presented in Supplementary Figs. 7 and 8.

**Evaluation of assumptions and sensitivity analyses.** The strength of the genetic instruments denoted by the F-statistic was ≥10 for all the accelerometer-measured physical activity variants and ranged between 27 and 56 (Table 2). Little evidence of directional pleiotropy was found for all models that used the extended 10 SNP instrument (MR-Egger intercept P-values > 0.06) (Table 1). The estimates from the weighted-median approach for the extended 10 SNP instrument were consistent with those of inverse-variance weighted (IVW) models (Table 1). The MR pleiotropy residual sum and outlier test (MR-PRESSO) method identified the SNPs rs11012732 and rs55657917 contained within the extended 10 SNP instrument as pleiotropic for breast cancer, but similar magnitude associations were observed when these variants were excluded from the analyses (Supplementary Table 10). After examining Phenoscanner and GWAS catalogue, we found that several of the accelerometer-measured physical activity genetic variants were also associated with adiposity-related phenotypes (Supplementary Tables 11, 12). However, the results from the leave-one-SNP out analysis did not reveal any influential SNPs driving the associations (Supplementary Tables 13–18). Additionally, similar results were found when the 5 adiposity-related SNPs were excluded from the extended 10 SNP genetic instrument (Supplementary Table 19). Further, the results from the multivariable MR analyses adjusting for BMI using the extended 10 SNP instrument were largely

**Table 1 Mendelian Randomisation estimates between accelerometer-measured physical activity and cancer risk.**

| Methods | | Genome-wide significant SNPs ($n = 5$) from the GWAS by Doherty et al.[11] | | | | | Extended number of SNPs ($n = 10$) from the GWAS by Klimentidis et al.[12] | | | | |
|---|---|---|---|---|---|---|---|---|---|---|---|
| | No. Cases | Estimates (OR)[a] | 95% CI | P-value | Q-value | P-value for pleiotropy[b] or heterogeneity[c] | Estimates (OR)[a] | 95% CI | P-value | Q-value | P-value for pleiotropy[b] or heterogeneity[c] |
| *Breast cancer* | | | | | | | | | | | |
| Inverse-variance weighted[d] | 122,977 | 0.51 | 0.27, 0.98 | 0.04 | 0.062 | $4.4 \times 10^{-8}$ | 0.59 | 0.42, 0.84 | 0.003 | 0.012 | $6.8 \times 10^{-7}$ |
| MR-Egger | | 0.01 | 0.00, 2.01 | 0.09 | | 0.16 | 0.55 | 0.09, 3.20 | 0.5 | | 0.9 |
| Weighted median | | 0.61 | 0.42, 0.87 | 0.006 | | | 0.76 | 0.59, 0.98 | 0.03 | | |
| *ER$^{+ve}$ subset* | | | | | | | | | | | |
| Inverse-variance weighted[d] | 69,501 | 0.45 | 0.20, 1.01 | 0.054 | 0.077 | $8.5 \times 10^{-9}$ | 0.53 | 0.35, 0.82 | 0.004 | 0.004 | $3.1 \times 10^{-7}$ |
| MR-Egger | | 0.03 | 0.00, 40 | 0.34 | | 0.46 | 0.61 | 0.07, 5.26 | 0.65 | | 0.9 |
| Weighted median | | 0.55 | 0.35, 0.85 | 0.008 | | | 0.66 | 0.48, 0.90 | 0.008 | | |
| *ER$^{-ve}$ subset* | | | | | | | | | | | |
| Inverse-variance weighted[d] | 21,468 | 0.95 | 0.44, 2.04 | 0.89 | 0.89 | 0.002 | 0.78 | 0.51, 1.22 | 0.27 | 0.3 | 0.01 |
| MR-Egger | | 0.01 | 0.00, 4.48 | 0.15 | | 0.15 | 0.24 | 0.03, 1.81 | 0.17 | | 0.24 |
| Weighted median | | 0.84 | 0.47, 1.47 | 0.53 | | | 0.7 | 0.47, 1.04 | 0.08 | | |
| *Colorectal cancer* | | | | | | | | | | | |
| Inverse-variance weighted | 52,775 | 0.66 | 0.48, 0.90 | 0.01 | 0.022 | 0.39 | 0.6 | 0.47, 0.76 | $2.4 \times 10^{-5}$ | 0.0002 | 0.5 |
| MR-Egger | | 0.32 | 0.01, 6.69 | 0.46 | | 0.64 | 0.24 | 0.08, 0.72 | 0.011 | | 0.1 |
| Weighted median | | 0.6 | 0.39, 0.92 | 0.02 | | | 0.61 | 0.44, 0.85 | 0.003 | | |
| *Colorectal cancer in men* | | | | | | | | | | | |
| Inverse-variance weighted | 28,207 | 0.79 | 0.50, 1.23 | 0.29 | 0.31 | 0.22 | 0.76 | 0.55, 1.07 | 0.11 | 0.14 | 0.62 |
| MR-Egger | | 16.4 | 0.32, 812 | 0.16 | | 0.13 | 0.59 | 0.12, 2.81 | 0.51 | | 0.74 |
| Weighted median | | 0.64 | 0.34, 1.19 | 0.16 | | | 0.8 | 0.51, 1.27 | 0.34 | | |
| *Colorectal cancer in women* | | | | | | | | | | | |
| Inverse-variance weighted | 24,568 | 0.57 | 0.36, 0.90 | 0.02 | 0.036 | 0.08 | 0.49 | 0.35, 0.68 | $3.0 \times 10^{-5}$ | 0.0002 | 0.19 |
| MR-Egger | | 0.01 | 0.00, 0.54 | 0.02 | | 0.045 | 0.11 | 0.02, 0.55 | 0.007 | | 0.06 |
| Weighted median | | 0.61 | 0.32, 1.16 | 0.13 | | | 0.47 | 0.29, 0.75 | 0.002 | | |
| *Colon cancer* | | | | | | | | | | | |
| Inverse-variance weighted | 27,817 | 0.64 | 0.44, 0.94 | 0.02 | 0.036 | 0.17 | 0.56 | 0.42, 0.73 | $4.4 \times 10^{-5}$ | 0.0002 | 0.57 |
| MR-Egger | | 0.42 | 0.00, 40.5 | 0.71 | | 0.86 | 0.35 | 0.09, 1.29 | 0.11 | | 0.47 |
| Weighted median | | 0.62 | 0.36, 1.06 | 0.08 | | | 0.49 | 0.34, 0.72 | $3.0 \times 10^{-4}$ | | |
| *Proximal colon cancer* | | | | | | | | | | | |
| Inverse-variance weighted | 12,360 | 0.66 | 0.41, 1.06 | 0.09 | 0.12 | 0.72 | 0.6 | 0.42, 0.86 | 0.005 | 0.014 | 0.9 |
| MR-Egger | | 0.62 | 0.01, 33.12 | 0.82 | | 0.98 | 0.33 | 0.06, 1.71 | 0.18 | | 0.46 |
| Weighted median | | 0.67 | 0.36, 1.22 | 0.19 | | | 0.56 | 0.35, 0.89 | 0.01 | | |
| *Distal colon cancer* | | | | | | | | | | | |
| Inverse-variance weighted | 14,016 | 0.51 | 0.31, 0.83 | 0.007 | 0.018 | 0.74 | 0.45 | 0.31, 0.64 | $1.7 \times 10^{-5}$ | 0.0002 | 0.72 |
| MR-Egger | | 0.32 | 0.00, 121 | 0.71 | | 0.88 | 0.34 | 0.06, 1.89 | 0.22 | | 0.75 |
| Weighted median | | 0.5 | 0.25, 1.00 | 0.051 | | | 0.45 | 0.28, 0.75 | 0.002 | | |
| *Rectal cancer* | | | | | | | | | | | |
| Inverse-variance weighted | 13,713 | 0.7 | 0.43, 1.14 | 0.15 | 0.18 | 0.13 | 0.68 | 0.47, 0.98 | 0.04 | 0.062 | 0.24 |
| MR-Egger | | 3.49 | 0.01, 1635 | 0.69 | | 0.6 | 0.43 | 0.06, 3.26 | 0.41 | | 0.65 |
| Weighted median | | 0.94 | 0.49, 1.79 | 0.85 | | | 0.76 | 0.47, 1.27 | 0.3 | | |

*CI* confidence intervals, *MR* Mendelian randomisation, *OR* odds ratio, *SNPs* Single nucleotide polymorphisms
[a]The estimates correspond to a standard deviation increase in physical activity
Q-value: False discovery rate (FDR) correction performed using the Benjamini–Hochberg method
[b]P-value or pleiotropy based on MR-Egger intercept
[c]P-value for heterogeneity based on Q statistic
[d]The estimates were derived from a random effects model due to the presence of heterogeneity based on Cochran's Q statistic

unchanged from the main IVW results (Supplementary Table 20). Finally, a similar pattern of results was found when GWAS effect estimates adjusted for BMI were used for 5 SNP genetic instrument[11] (Supplementary Table 21).

## Discussion

In this MR analysis, higher levels of genetically predicted accelerometer-measured physical activity were associated with lower risks of breast cancer and colorectal cancer, with similar magnitude inverse associations found for ER$^{+ve}$ and for colon cancer. These findings indicate that population-level increases in physical activity may lower the incidence of these two commonly diagnosed cancers, and support the promotion of physical activity for cancer prevention.

A large body of observational studies has investigated how physical activity relates to risk of breast and colorectal cancer[15,16].

In a participant-level pooled analysis of 12 prospective studies, when the 90th and 10th percentile of leisure-time physical activity were compared, lower risks of breast cancer (hazard ratio [HR]: 0.90, 95% CI: 0.87 to 0.93), colon cancer (HR: 0.84, 95% CI: 0.77 to 0.91), and rectal cancer (HR: 0.87, 95% CI: 0.80 to 0.95) were found[3]. Similarly, inverse associations between total physical activity and risks of postmenopausal breast and colorectal cancer were recently reported in meta-analyses of all published prospective cohort data by the WCRF/AICR Continuous Update Project[15,16].

These observational studies relied on self-report physical activity assessment methods that are prone to measurement error, which may attenuate associations towards the null. In addition, causality cannot be ascertained from such observational analyses as they are vulnerable to residual confounding and reverse causality. Further, logistical and financial challenges prohibit

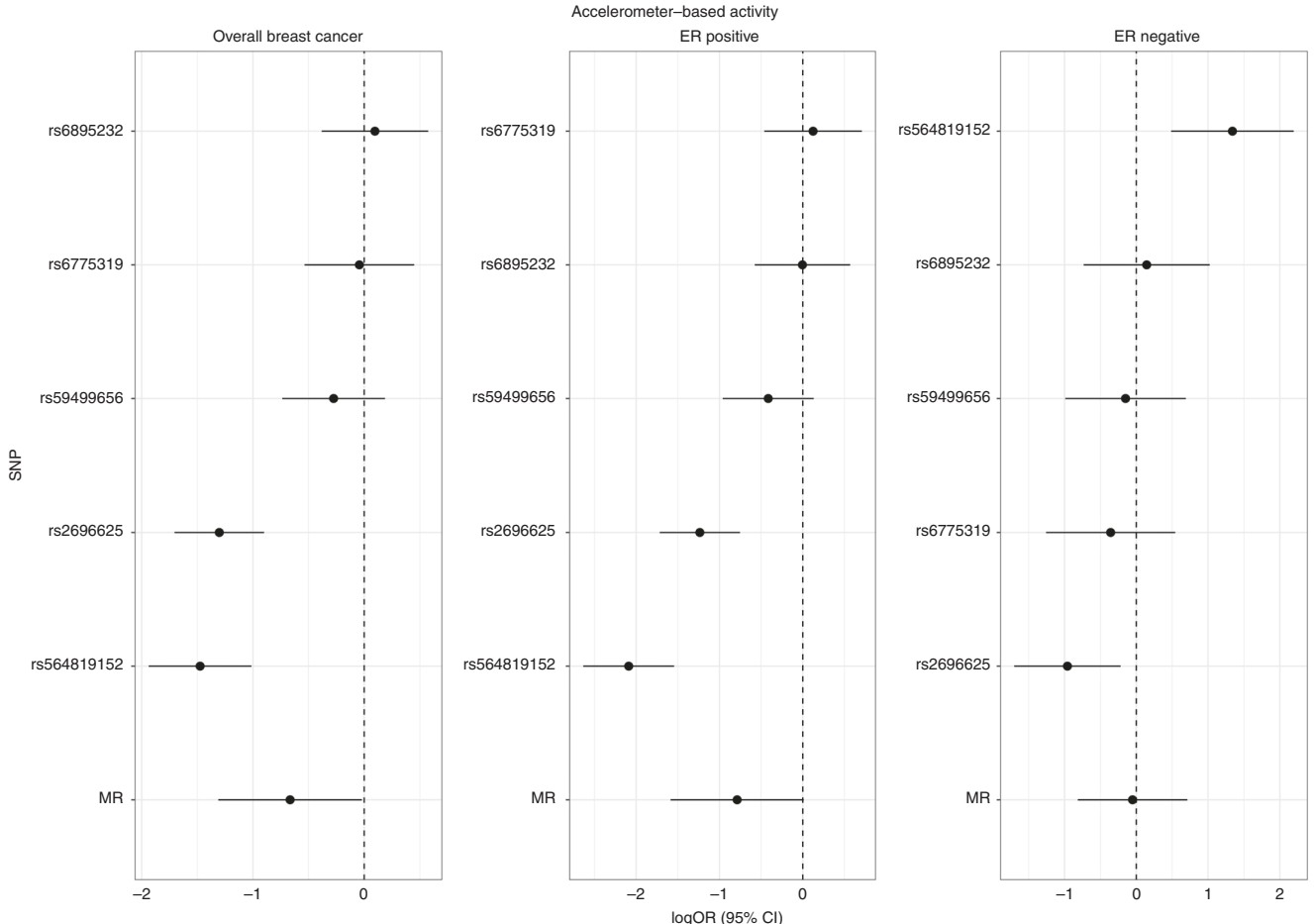

**Fig. 1 Mendelian randomisation analysis for individual SNPs associated with accelerometer-measured physical activity in relation to breast cancer risk using the genetic instrument from the GWAS by Doherty et al.[11].** The x axis corresponds to a log OR per one unit increase in the physical activity based on the average acceleration (milligravities). The Mendelian randomisation (MR) result corresponds to a random effects model due to heterogeneity across the genetic instruments. logOR = log odds ratio (black filled circle). 95% CI = 95% confidence interval (black line). SNP single nucleotide polymorphism.

randomised controlled trials of physical activity and cancer development. For example, it has been estimated that in order to detect a 20% breast cancer risk reduction, between 26,000 to 36,000 healthy middle-aged women would need to be randomised to a 5 year exercise intervention[17]. Several trials on cancer survivors are registered and underway, and these may provide evidence of potential causal associations between physical activity and disease free survival and cancer recurrence;[18] however, these interventions will not inform causal inference of the relationship between physical activity and cancer development. We therefore conducted MR analyses to allow causal inference between accelerometer-measured physical activity and risks of developing breast and colorectal cancer. The inverse associations we found were stronger for ER+ve breast cancer and colon cancer, and are highly concordant with prior observational epidemiological evidence.

There is currently no standard method in translating accelerometer data into energy expenditure values, such as metabolic equivalent of tasks (METs). However, using an accepted threshold for moderate activity (e.g. fast walking) of 100 milligravity[19,20], 1-SD higher mean acceleration (~8 milli-gravity) equates to approximately 50 min extra moderate activity per week. Similarly, using an accepted threshold of 425 milli-gravity for vigorous activity (e.g. running)[19,20], a 1-SD higher mean acceleration equates to approximately 8 min of extra vigorous activity per week. In our study, we found that such an increase in

weekly activity translates to a 49 and 34% lower risks of developing breast and colorectal cancer, respectively.

Being physically active is associated with less weight gain and body fatness, and lower adiposity is associated with lower risks of breast and colorectal cancer[15,16]. Since body size/adiposity is likely on the causal pathway linking physical activity and breast and colorectal cancer, it is challenging to disentangle independent effects of physical activity on cancer development. The close inter-relation between adiposity and physical activity is evident from 5 of the 10 SNPs in the extended genetic instrument for accelerometer-measured physical activity being previously associated with adiposity/body size traits. However, it is noteworthy that our results were unchanged when we excluded adiposity-related SNPs from this genetic instrument, and when we conducted multivariable MR analyses adjusting for body mass index (BMI). These results would therefore suggest that physical activity is also associated with breast and colorectal cancer independently of adiposity.

Multiple biological mechanisms are hypothesised to mediate the potential beneficial role of physical activity on cancer development[21,22]. Greater physical activity has been associated with lower circulating levels of insulin and insulin-like growth factors, which promote cellular proliferation in breast and colorectal tissue and have also been linked to development of cancers at these sites[21,23–27]. Higher levels of physical activity have also been associated with lower circulating concentrations of estradiol,

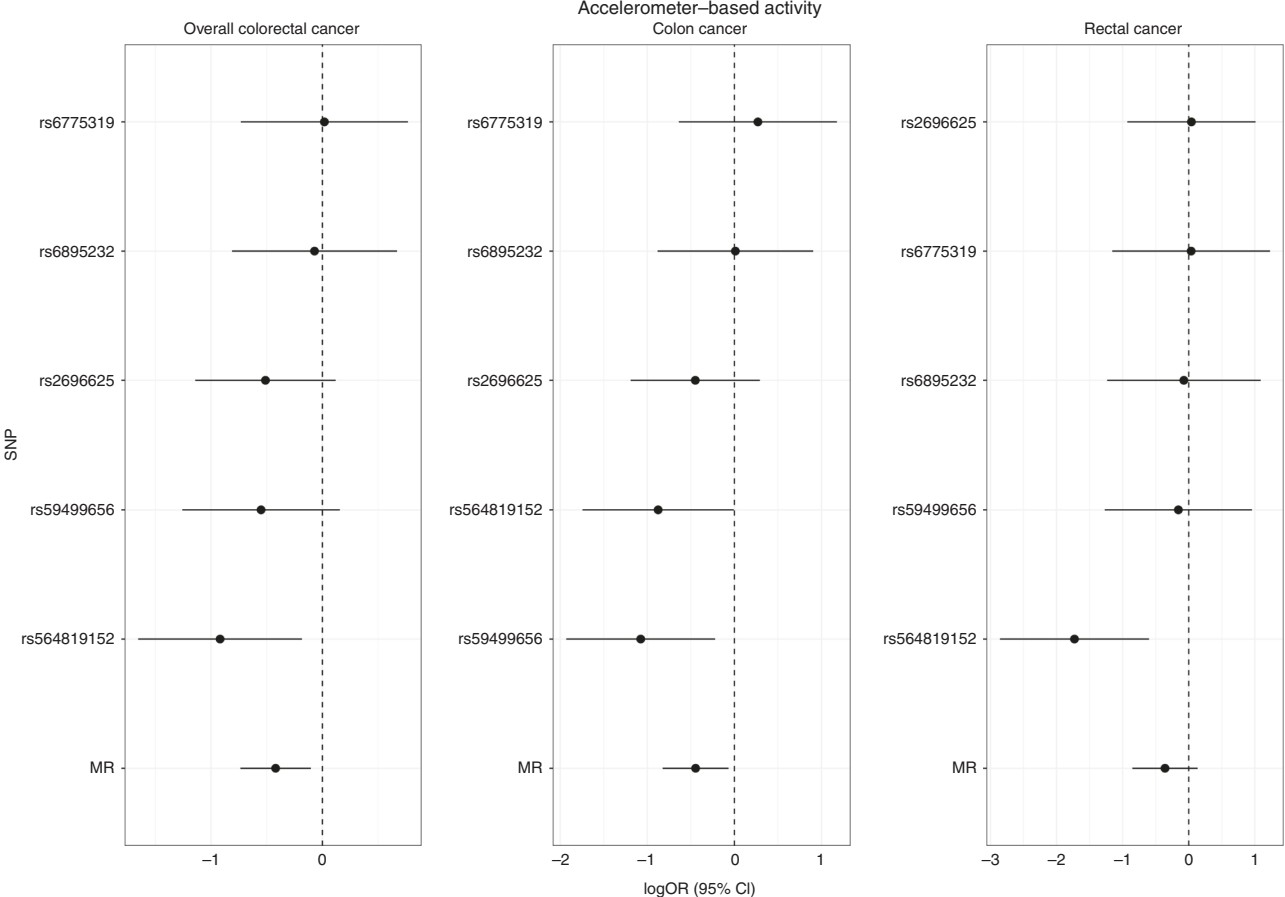

**Fig. 2 Mendelian randomisation analysis for individual SNPs associated with accelerometer-measured physical activity in relation to colorectal cancer risk (overall, colon, rectal) using the genetic instrument from the GWAS by Doherty et al.[11].** The x axis corresponds to a log OR per one unit increase in the physical activity based on the average acceleration (milli-gravities). The Mendelian randomisation (MR) result corresponds to a random effects model due to heterogeneity across the genetic instruments. logOR = log odds ratio (black filled circle). 95% CI = 95% confidence interval (black line). SNP single nucleotide polymorphism.

estrone, and higher levels of sex hormone binding globulin[28–30] which are themselves risk factors for breast cancer development[31,32]. Physical activity has also been associated with improvements in the immune response with increased surveillance and elimination of cancerous cells[33,34]. Higher levels of physical activity may also reduce systemic inflammation by lowering the levels of pro-inflammatory factors, such as C-reactive protein (CRP), interleukin-6 (IL-6) and tumour necrosis factor-alpha (TNF-a)[33,35,36]. Finally, emerging evidence suggests that the gut microbiome may play an important role in the physical activity and cancer relationship. Dysbiosis of the gut microbiome has been associated with increased risks of several malignancies, including breast and colorectal cancer[37]. Changes in gut microbiome composition and derived metabolic products have been found following endurance exercise training with short-chain fatty acid concentrations increased in lean, but not obese, subjects[38,39].

A fundamental assumption of MR is that the genetic variants do not influence the outcome via a different biological pathway from the exposure of interest (horizontal pleiotropy). We conducted multiple sensitivity analyses using an extended 10 SNP genetic instrument for accelerometer-measured physical activity to test for the influence of pleiotropy on our causal estimates, and our results were robust according to these various tests. A potential limitation of our analysis is that the genetic variants explained a small fraction of the variability of accelerometer-

measured physical activity, which may have resulted in some of the breast cancer subtype and colorectal subsite analyses being underpowered. In addition, our use of summary-level data precluded subgroup analyses by other cancer risk factors (e.g. BMI, exogenous hormone use). We were also unable to stratify breast cancer analyses by menopausal status; however, the majority of women in the source GWAS had postmenopausal breast cancer[13]. Finally, 7-day accelerometer-measured physical activity levels of UK Biobank participants may not have been representative of usual behavioural patterns.

In conclusion, we found that genetically elevated levels of accelerometer-measured physical activity were associated with lower risks of breast and colorectal cancer. These findings strongly support the promotion of physical activity as an effective strategy in the primary prevention of these commonly diagnosed cancers.

## Methods

**Data on physical activity**. Summary-level data were obtained from two recently published GWAS on accelerometer-measured physical activity conducted in ~91,000 participants from the UK Biobank[11,12]. In the GWAS by Doherty et al.[11], BOLT-LMM was used to perform linear mixed models analyses that were adjusted for assessment centre, genotyping array, age, age[2], and season. This GWAS identified 5 genome-wide-significant SNPs (P-value $< 5 \times 10^{-8}$) associated with accelerometer-measured physical activity. The estimated SNP-based heritability for accelerometer-measured physical activity in the UK Biobank is 14%[12], suggesting that additional SNPs contributed to its variation. Consequently, we also used an accelerometer-measured physical activity instrument with an expanded number

**Table 2 Summary information on accelerometer-measured physical activity SNPs used as genetic instruments used for the Mendelian randomisation analyses.**

| SNP | Effect allele | Baseline allele | Chr | Position[a] | Gene | EAF | beta PA[b] | se PA | N[c] | $R^2$ | F-statistic |
|---|---|---|---|---|---|---|---|---|---|---|---|
| *5 SNPs from GWAS by Doherty et al. 2018[11]* | | | | | | | | | | | |
| rs6775319 | A | T | 3 | 18717009 | SATB1-AS1 | 0.27 | 0.03 | 0.005 | 91,105 | 0.0003 | 27 |
| rs6895232 | T | A | 5 | 152659861 | LINC01470 | 0.66 | 0.03 | 0.005 | 91,105 | 0.0003 | 30 |
| rs564819152 | A | G | 10 | 21531721 | SKIDA1 | 0.68 | 0.03 | 0.005 | 91,105 | 0.0003 | 31 |
| rs2696625 | G | A | 17 | 46249498 | KANSL1-AS1 | 0.23 | 0.04 | 0.005 | 91,105 | 0.0005 | 44 |
| rs59499656 | T | A | 18 | 43188344 | RIT2/SYT4 | 0.35 | 0.03 | 0.005 | 91,105 | 0.0004 | 32 |
| *10 SNPs from GWAS by Klimentidis et al. 2018[12]* | | | | | | | | | | | |
| rs12045968 | G | T | 1 | 33225097 | ZNF362 | 0.22 | 0.24 | 0.044 | 91,084 | 0.0003 | 30 |
| rs34517439 | C | A | 1 | 77984833 | DNAJB4 | 0.91 | 0.31 | 0.056 | 91,084 | 0.0003 | 30 |
| rs6775319 | A | T | 3 | 18717009 | LOC105376976 | 0.3 | 0.23 | 0.041 | 91,084 | 0.0003 | 30 |
| rs12522261 | G | A | 5 | 152675265 | LINC01470 | 0.67 | 0.21 | 0.038 | 91,084 | 0.0003 | 31 |
| rs9293503 | T | C | 5 | 88653144 | LINC00461 | 0.88 | 0.33 | 0.059 | 91,084 | 0.0003 | 31 |
| rs11012732 | A | G | 10 | 21541175 | MLLT10 | 0.65 | 0.23 | 0.039 | 91,084 | 0.0004 | 33 |
| rs148193266 | C | A | 11 | 104657953 | RP11-681H10.1 | 0.02 | 0.51 | 0.092 | 91,084 | 0.0003 | 31 |
| rs1550435 | T | C | 15 | 74039044 | PML | 0.53 | 0.2 | 0.037 | 91,084 | 0.0003 | 29 |
| rs55657917 | G | T | 17 | 45767194 | CRHR1 | 0.22 | 0.3 | 0.04 | 91,084 | 0.0006 | 56 |
| rs59499656 | T | A | 18 | 43188344 | RIT2/SYT4 | 0.34 | 0.23 | 0.038 | 91,084 | 0.0004 | 36 |

*BMI* body mass index, *Chr* chromosome, *EAF* effect allele frequency, *NA* not available, *PA* physical activity, *se* standard error, *SNP* single nucleotide polymorphism
[a]Position based on GRCh38.p12
[b]The beta coefficients are expressed in milligravities
[c]N refers to the sample size of the initial GWAS from which the genetic variants were selected

of SNPs ($n = 10$; associated with accelerometer-measured physical activity at P-value $< 1 \times 10^{-7}$) identified by another UK Biobank GWAS by Klimentidis et al.[12]. The extended number of SNPs in the accelerometer-measured physical activity instrument allowed us to conduct more robust sensitivity analyses to check for the influence of horizontal pleiotropy on the results. Data for the associations between the 10 SNPs and physical activity were obtained from a recent MR study on physical activity and depression that used the data from the same UK Biobank GWAS[40]. Detailed information on the genetic variants used in the 5 genome-wide significant SNP instrument and the extended 10 SNP instrument is provided in Table 2.

**Data on breast cancer and colorectal cancer.** Summary data for the associations of the accelerometer-measured genetic variants with breast cancer (overall and by estrogen receptor status: ER positive [ER+ve] and ER negative [ER-ve]) were obtained from a GWAS of 228,951 women (122,977 breast cancer [69,501 ER positive, 21,468 ER negative] cases and 105,974 controls) of European ancestry from the Breast Cancer Association Consortium (BCAC)[13]. Genotyping data were imputed with the program IMPUTE2[14] with the 1000 Genomes Project Phase III integrated variant set as the reference panel. Single nucleotide polymorphisms (SNPs) with low imputation quality (imputation $r^2 < 0.5$) were excluded. Top principal components (PCs) were included as covariates in regression analysis to address potential population substructure (iCOGS: top eight PCs; OncoArray: top 15 PCs) (Supplementary Tables 1, 2)[13,41]. For colorectal cancer, summary data from 98,715 participants (52,775 colorectal cancer cases and 45,940 controls) were drawn from a meta-analysis within the ColoRectal Transdisciplinary Study (CORECT), the Colon Cancer Family Registry (CCFR), and the Genetics and Epidemiology of Colorectal Cancer (GECCO) consortia[14]. Imputation was performed using the Haplotype Reference Consortium (HRC) r1.0 reference panel and the regression models were further adjusted for age, sex, genotyping platform (whenever appropriate), and genomic PCs (from 3 to 13, whenever appropriate) (Supplementary Tables 3–6).

**Statistical power.** The a priori statistical power was calculated using an online tool at http://cnsgenomics.com/shiny/mRnd/[42]. The 5 and 10 SNP accelerometer-measured physical activity instruments explained an estimated 0.2% and 0.4% of phenotypic variability, respectively. Given a type 1 error of 5%, for the 5 SNP instrument identified from the GWAS by Doherty et al.[11] we had sufficient power (> 80%) when the expected OR per 1 SD was ≤ 0.77 and ≤ 0.67 for overall breast cancer (122,977 cases and 105,974 controls) and colorectal cancer (52,775 colorectal cancer cases and 45,940 controls), respectively. Power estimates for the 5 genome-wide significant SNP and the extended 10 SNP instruments by subtypes of breast cancer and subsites of colorectal cancer are presented in Supplementary Tables 7 and 8.

**Statistical analysis.** A two-sample MR approach using summary data and the fixed-effect IVW method was implemented. All accelerometer-measured physical activity and cancer results correspond to an OR per 1 SD increment (8.14 milli-gravities) in the genetically predicted overall average acceleration. The

heterogeneity of causal effects by cancer subtype and sex was investigated by estimating the $I^2$ statistic assuming a fixed-effects model[43].

For causal estimates from MR studies to be valid, three main assumptions must be met: 1) the genetic instrument is strongly associated with the level of accelerometer-measured physical activity; 2) the genetic instrument is not associated with any potential confounder of the physical activity—cancer association; and 3) the genetic instrument does not affect cancer independently of physical activity (i.e. horizontal pleiotropy should not be present)[44]. The strength of each instrument was measured by calculating the F-statistic using the following formula: $F = R^2(N - 2)/(1 - R^2)$, where $R^2$ is the proportion of the variability of the physical activity explained by each instrument and N the sample size of the GWAS for the SNP-physical activity association[45]. To calculate $R^2$ for the 5 genome-wide significant SNP instrument we used the following formula: $2 \times EAF \times (1 - EAF) \times beta^2$; whereas for the extended 10 SNP instrument we used: $(2 \times EAF \times (1 - EAF) \times beta^2)/[(2 \times EAF \times (1 - EAF) \times beta^2) + (2 \times EAF \times (1 - EAF) \times N \times SE(beta)^2)]$, where EAF is the effect allele frequency, beta is the estimated genetic effect on physical activity, N is the sample size of the GWAS for the SNP-physical activity association and SE (beta) is the standard error of the genetic effect[46]. FDR correction (Q-value) was performed using the Benjamini–Hochberg method[47].

**Sensitivity analyses.** Several sensitivity analyses were used to check and correct for the presence of pleiotropy in the causal estimates. Cochran's Q was computed to quantify heterogeneity across the individual causal effects, with a P-value ≤ 0.05 indicating the presence of pleiotropy, and that consequently, a random effects IVW MR analysis should be used[43,48]. We also assessed the potential presence of horizontal pleiotropy using MR-Egger regression based on its intercept term, where deviation from zero denotes the presence of directional pleiotropy. Additionally, the slope of the MR-Egger regression provides valid MR estimates in the presence of horizontal pleiotropy when the pleiotropic effects of the genetic variants are independent from the genetic associations with the exposure[49,50]. We also computed OR estimates using the complementary weighted-median method that can give valid MR estimates under the presence of horizontal pleiotropy when up to 50% of the included instruments are invalid[44]. The presence of pleiotropy was also assessed using the MR-PRESSO. In this, outlying SNPs are excluded from the accelerometer-measured physical activity instrument and the effect estimates are reassessed[51]. For all of the aforementioned sensitivity analyses to identify possible pleiotropy, we considered the estimates from the extended 10 SNP instrument as the primary results due to unstable estimates from the 5 SNP instrument. A leave-one-SNP out analysis was also conducted to assess the influence of individual variants on the observed associations. We also examined the selected genetic instruments and their proxies ($r^2 > 0.8$) and their associations with secondary phenotypes (P-value $< 5 \times 10^{-8}$) in Phenoscanner (http://www.phenoscanner. medschl.cam.ac.uk/) and GWAS catalog (date checked April 2019).

For the extended 10 SNP instrument, we also conducted multivariable MR analyses to adjust for potential pleiotropy due to BMI because the initial GWAS on physical activity reported several strong associations (P-value $< 10^{-5}$) between the identified SNPs and BMI[52]. The new estimates correspond to the direct causal effect of physical activity with the BMI being fixed. The genetic data on BMI were

obtained from a GWAS study published by The Genetic Investigation of ANthropometric Traits (GIANT) consortium[53] (Supplementary Table 9). Additionally, for the extended 10 SNP instrument, we also conducted analyses with adiposity-related SNPs (i.e. those previously associated with BMI, waist circumference, weight, or body/trunk fat percentage in GWAS studies at $P$-value < $10^{-8}$) excluded ($n = 5$; rs34517439, rs6775319, rs11012732, rs1550435, rs59499656). Finally, we conducted two-sample MR analyses using BMI adjusted GWAS estimates for the 5 SNP accelerometer-measured physical activity instrument[11]. However, the MR results using the BMI adjusted GWAS estimates should be interpreted cautiously due to the potential for collider bias[11].

All the analyses were conducted using the MendelianRandomisation[54] and TwoSampleMR[55] packages, and the R programming language.

**Reporting summary**. Further information on research design is available in the Nature Research Reporting Summary linked to this article.

## Data availability
Data supporting the findings of this study are available within the paper and its supplementary information files.

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

## Acknowledgements

This work was supported by the National Cancer Institute, the International Agency for Research on Cancer and a Cancer Research UK program grant (C18281/A19169 to RMM, SJL & NK). RMM was supported by the National Institute for Health Research (NIHR) Bristol Biomedical Research Centre. The views expressed are those of the author (s) and not necessarily those of the NIHR or the Department of Health and Social Care. The funding sources for BCAC, CCFR, GECCO, and CORECT consortia are presented in detail in the appendix in the Supplementary material.

## Author contributions

Study conception: M.J.G. and N.M. Data analysis: N.P. and N.M. Drafting of the manuscript: N.P., M.J.G., and N.M. All other authors (N.D., K.K.T., B.B., R.M.M., S.J.L., N.K., T.M.R., D.A., K.A., S.I.B., D.T.B., H.B., D.B.B., B.B.-d.-M., P.T.C., S.C.B., A.T.C., J.C.C., M.E.D., J.C.F., S.J.G., G.G.G., E.G., S.B.G., A.G., J.H., H.H., S.H., T.A.H., M.H., J.L.H., L.H., J.M.H., J.R.H., M.A.J., T.O.K., T.K., C.L.V., L.L.M., C.I.L., L.L., A.L., N.M.L., B.L., S.D.M., G.M., A.M.M., R.M., E.M., L.M., V.M., P.A.N., K.O., V.P., P.D.P.P., E.A.P., J.D.P., G.R., E.R., M.J.S., S.L.S., R.E.S., G.S., S.S., M.L.S., M.S., C.M.T., S.N.T., R.C.T., A.T., C.M.U., .F.J.B.v.D., B.V.G., P.V., E.W., A.W., M.O.W., A.H.W., U.P.) contributed to the interpretation of the results and critical revision of the manuscript.

## Competing interests

Where authors are identified as personnel of the International Agency for Research on Cancer/World Health Organization, the authors alone are responsible for the views expressed in this article and they do not necessarily represent the decisions, policy or views of the International Agency for Research on Cancer / World Health Organization. The authors declare no competing interests.

## Additional information

Nikos Papadimitriou[1], Niki Dimou[1], Konstantinos K. Tsilidis[2,3], Barbara Banbury[4], Richard M. Martin[5,6,7], Sarah J. Lewis[6], Nabila Kazmi[5], Timothy M. Robinson[6], Demetrius Albanes[8], Krasimira Aleksandrova[9], Sonja I. Berndt[8], D. Timothy Bishop[10], Hermann Brenner[11,12,13], Daniel D. Buchanan[14,15,16], Bas Bueno-de-Mesquita[17,18,19,20], Peter T. Campbell[21], Sergi Castellví-Bel[22], Andrew T. Chan[23,24], Jenny Chang-Claude[25,26], Merete Ellingjord-Dale[3], Jane C. Figueiredo[27,28], Steven J. Gallinger[29], Graham G. Giles[14,30], Edward Giovannucci[31,32,33], Stephen B. Gruber[34], Andrea Gsur[35], Jochen Hampe[36], Heather Hampel[37], Sophia Harlid[38], Tabitha A. Harrison[4], Michael Hoffmeister[11], John L. Hopper[14,39], Li Hsu[4,40], José María Huerta[41,42], Jeroen R. Huyghe[4], Mark A. Jenkins[14], Temitope O. Keku[43], Tilman Kühn[25], Carlo La Vecchia[44,45], Loic Le Marchand[46], Christopher I. Li[4], Li Li[47], Annika Lindblom[48,49], Noralane M. Lindor[50], Brigid Lynch[14,30,51], Sanford D. Markowitz[52], Giovanna Masala[53], Anne M. May[54], Roger Milne[14,30,55], Evelyn Monninkhof[54], Lorena Moreno[22], Victor Moreno[41,56,57], Polly A. Newcomb[4,58], Kenneth Offit[59,60], Vittorio Perduca[61,62,63], Paul D.P. Pharoah[64], Elizabeth A. Platz[65], John D. Potter[4], Gad Rennert[66,67,68], Elio Riboli[3], Maria-Jose Sánchez[41,69], Stephanie L. Schmit[34,70], Robert E. Schoen[71], Gianluca Severi[61,62], Sabina Sieri[72], Martha L. Slattery[73], Mingyang Song[23,24,31,32], Catherine M. Tangen[74], Stephen N. Thibodeau[75], Ruth C. Travis[76], Antonia Trichopoulou[44], Cornelia M. Ulrich[77], Franzel J.B. van Duijnhoven[78], Bethany Van Guelpen[79,80],

Pavel Vodicka[81,82,83], Emily White[4,84], Alicja Wolk [85], Michael O. Woods[86], Anna H. Wu[87], Ulrike Peters [4,84], Marc J. Gunter[1,88] & Neil Murphy [1,88]*

[1]Section of Nutrition and Metabolism, International Agency for Research on Cancer, Lyon, France. [2]Department of Hygiene and Epidemiology, University of Ioannina School of Medicine, Ioannina, Greece. [3]Department of Epidemiology and Biostatistics, School of Public Health, Imperial College London, London, UK. [4]Public Health Sciences Division, Fred Hutchinson Cancer Research Center, Seattle, WA, USA. [5]MRC Integrative Epidemiology Unit (IEU), Population Health Sciences, Bristol Medical School, University of Bristol, Bristol, UK. [6]Bristol Medical School, Department of Population Health Sciences, University of Bristol, Bristol, UK. [7]National Institute for Health Research (NIHR) Bristol Biomedical Research Centre, University Hospitals Bristol NHS Foundation Trust and the University of Bristol, Bristol, UK. [8]Division of Cancer Epidemiology and Genetics, National Cancer Institute, National Institutes of Health, Bethesda, MA, USA. [9]German Institute of Human Nutrition Potsdam-Rehbruecke (DIfE), Arthur-Scheunert-Allee 114-116, 14558 Nuthetal, Germany. [10]Leeds Institute of Cancer and Pathology, University of Leeds, Leeds, UK. [11]Division of Clinical Epidemiology and Aging Research, German Cancer Research Center (DKFZ), Heidelberg, Germany. [12]Division of Preventive Oncology, German Cancer Research Center (DKFZ) and National Center for Tumor Diseases (NCT), Heidelberg, Germany. [13]German Cancer Consortium (DKTK), German Cancer Research Center (DKFZ), Heidelberg, Germany. [14]Centre for Epidemiology and Biostatistics, Melbourne School of Population and Global Health, The University of Melbourne, Melbourne, VIC, Australia. [15]Colorectal Oncogenomics Group, Genetic Epidemiology Laboratory, Department of Pathology, The University of Melbourne, Parkville, VIC, Australia. [16]Genetic Medicine and Family Cancer Clinic, The Royal Melbourne Hospital, Parkville, VIC, Australia. [17]Former senior scientist, Dept. for Determinants of Chronic Diseases (DCD), National Institute for Public Health and the Environment (RIVM), PO Box 1, 3720 BA Bilthoven, Netherlands. [18]Former associate professor, Department of Gastroenterology and Hepatology, University Medical Centre, Utrecht, Netherlands. [19]Former visiting professor, Dept. of Epidemiology and Biostatistics, The School of Public Health, Imperial College London, St Mary's Campus, Norfolk Place, London, W2 1PG London, UK. [20]Former academic Icon / visiting professor, Dept. of Social & Preventive Medicine, Faculty of Medicine, University of Malaya, Pantai Valley, 50603 Kuala Lumpur, Malaysia. [21]Behavioral and Epidemiology Research Group, American Cancer Society, Atlanta, GA, USA. [22]Gastroenterology Department, Hospital Clínic, Institut d'Investigacions Biomèdiques August Pi i Sunyer (IDIBAPS), Centro de Investigación Biomédica en Red de Enfermedades Hepáticas y Digestivas (CIBEREHD), University of Barcelona, Barcelona, Spain. [23]Division of Gastroenterology, Massachusetts General Hospital and Harvard Medical School, Boston, MA, USA. [24]Clinical and Translational Epidemiology Unit, Massachusetts General Hospital and Harvard Medical School, Boston, MA, USA. [25]Division of Cancer Epidemiology, German Cancer Research Center (DKFZ), Heidelberg, Germany. [26]University Medical Centre Hamburg-Eppendorf, University Cancer Centre Hamburg (UCCH), Hamburg, Germany. [27]Department of Medicine, Samuel Oschin Comprehensive Cancer Institute, Cedars-Sinai Medical Center, Los Angeles, CA, USA. [28]Department of Preventive Medicine, Keck School of Medicine, University of Southern California, Los Angeles, CA, USA. [29]Lunenfeld Tanenbaum Research Institute, Mount Sinai Hospital, University of Toronto, Toronto, ON, Canada. [30]Cancer Epidemiology and Intelligence Division, Cancer Council Victoria, Melbourne, VIC, Australia. [31]Department of Epidemiology, Harvard T.H. Chan School of Public Health, Harvard University, Boston, MA, USA. [32]Department of Nutrition, T.H. H, Chan School of Public Health, Boston, MA, USA. [33]Channing Division of Network Medicine, Brigham and Women's Hospital and Harvard Medical School, Boston, MA, USA. [34]Department of Preventive Medicine, USC Norris Comprehensive Cancer Center, Keck School of Medicine, University of Southern California, Los Angeles, CA, USA. [35]Institute of Cancer Research, Department of Medicine I, Medical University Vienna, Vienna, Austria. [36]Department of Medicine I, University Hospital Dresden, Technische Universität Dresden (TU Dresden), Dresden, Germany. [37]Division of Human Genetics, Department of Internal Medicine, The Ohio State University Comprehensive Cancer Center, Columbus, OH, USA. [38]Department of Radiation Sciences, Oncology, Umea University, 901 87 Umea, Sweden. [39]Department of Epidemiology, School of Public Health and Institute of Health and Environment, Seoul National University, Seoul, South Korea. [40]Department of Biostatistics, University of Washington, Seattle, WA, USA. [41]CIBER de Epidemiología y Salud Pública (CIBERESP), Madrid, Spain. [42]Department of Epidemiology, Murcia Regional Health Council, IMIB-Arrixaca, Murcia, Spain. [43]Center for Gastrointestinal Biology and Disease, University of North Carolina, Chapel Hill, NC, USA. [44]Hellenic Health Foundation, Athens, Greece. [45]Dept. of Clinical Sciences and Community Health, Università degli Studi di Milano, Milano, Italy. [46]University of Hawaii Cancer Center, Honolulu, HI, USA. [47]Department of Family Medicine, University of Virginia, Charlottesville, VA, USA. [48]Department of Clinical Genetics, Karolinska University Hospital, Stockholm, Sweden. [49]Department of Molecular Medicine and Surgery, Karolinska Institutet, Stockholm, Sweden. [50]Department of Health Science Research, Mayo Clinic, Scottsdale, AZ, USA. [51]Physical Activity Laboratory, Baker Heart and Diabetes Institute, Melbourne, VIC, Australia. [52]Departments of Medicine and Genetics, Case Comprehensive Cancer Center, Case Western Reserve University, and University Hospitals of Cleveland, Cleveland, OH, USA. [53]Cancer Risk Factors and Life-Style Epidemiology Unit, Institute for Cancer Research, Prevention and Clinical Network - ISPRO, Florence, Italy. [54]Julius Center for Health Sciences and Primary Care, University Medical Center Utrecht, Utrecht University, P.O. Box 85500, 3508 GA UTRECHT, Netherlands. [55]Genetic Epidemiology Laboratory, Department of Pathology, The University of Melbourne, Parkville, VIC, Australia. [56]Cancer Prevention and Control Program, Catalan Institute of Oncology-IDIBELL, L'Hospitalet de Llobregat, Barcelona, Spain. [57]Department of Clinical Sciences, Faculty of Medicine, University of Barcelona, Barcelona, Spain. [58]School of Public Health, University of Washington, Seattle, WA, USA. [59]Clinical Genetics Service, Department of Medicine, Memorial Sloan-Kettering Cancer Center, New York, NY, USA. [60]Department of Medicine, Weill Cornell Medical College, New York, NY, USA. [61]CESP, Fac. de médecine - Univ. Paris-Sud, Fac. de médecine - UVSQ I, Université Paris-Saclay, 94805 Villejuif, France. [62]Gustave Roussy, F-94805 Villejuif, France. [63]Laboratoire de Mathématiques Appliquées MAP5 (UMR CNRS 8145), Université Paris Descartes, Paris, France. [64]Department of Public Health and Primary Care, University of Cambridge, Cambridge, UK. [65]Department of Epidemiology, Johns Hopkins Bloomberg School of Public Health, Baltimore, MD, USA. [66]Department of Community Medicine and Epidemiology, Lady Davis Carmel Medical Center, Haifa, Israel. [67]Ruth and Bruce Rappaport Faculty of Medicine, Technion-Israel Institute of Technology, Haifa, Israel. [68]Clalit National Cancer Control Center, Haifa, Israel. [69]Andalusian School of Public Health, Biomedical Research Institute ibs.GRANADA, University of Granada, Granada, Spain. [70]Department of Cancer Epidemiology, H. Lee Moffitt Cancer Center and Research Institute, Tampa, FL, USA. [71]Department of Medicine and Epidemiology, University of Pittsburgh Medical Center, Pittsburgh, PA, USA. [72]Epidemiology and Prevention Unit, Fondazione IRCCS Istituto Nazionale dei Tumori, Milan, Italy. [73]Department of Internal Medicine, University of Utah, Salt Lake City, UT, USA. [74]SWOG Statistical Center, Fred Hutchinson Cancer Research Center, Seattle, WA, USA. [75]Division of Laboratory Genetics, Department of Laboratory Medicine and Pathology, Mayo Clinic, Rochester, MN, USA. [76]Cancer Epidemiology Unit, Nuffield Department of Population Health, University of Oxford, OX3 7LF Oxford, UK. [77]Huntsman Cancer Institute and Department of Population Health Sciences, University of Utah, Salt Lake City, UT, USA. [78]Division of Human Nutrition, Wageningen University and Research, Wageningen, Netherlands. [79]Department of Radiation Sciences, Oncology Unit, Umeå University, Umeå, Sweden. [80]Wallenberg Centre for Molecular Medicine, Umeå University, Umeå, Sweden. [81]Department of Molecular Biology of Cancer, Institute of Experimental Medicine of the Czech Academy of Sciences, Prague, Czech Republic. [82]Faculty of Medicine and Biomedical Center in Pilsen, Charles University, Pilsen, Czech Republic. [83]Institute of Biology and Medical Genetics, First Faculty of Medicine, Charles University, Prague, Czech Republic. [84]Department of

Epidemiology, University of Washington, Seattle, WA, USA. [85]Institute of Environmental Medicine, Karolinska Institutet, Stockholm, Sweden. [86]Memorial University of Newfoundland, Discipline of Genetics, St. John's, Canada. [87]University of Southern California, Preventative Medicine, Los Angeles, CA, USA. [88]These authors contributed equally: Marc J. Gunter, Neil Murphy. *email: MurphyN@iarc.fr

