## [Peer Review File · Nature Communications]

Reviewers' Comments:

Reviewer #1:

Remarks to the Author:

General Comments:

This large group of authors have written a novel and interesting paper on the association between physical activity and risks of breast and colorectal cancers using a Mendelian randomization (MR) analytic approach. This study was possible because they used the UK Biobank and GWA consortia data that included hundreds of thousands of study participants' physical activity and cancer incidence data. These types of large data sets have not been previously used to assess this particular research question. The major rationale for this analysis is that MR should overcome the confounding that exists in traditional analyses of epidemiologic studies that have relied on self-reported exposure data. Hence, there is interest in assessing if the associations observed in observational epidemiologic studies that have used self-reported data on physical activity and cancer incidence are also found in these MR analyses that should be free of these common biases. The authors conclude that there is support for a causal association between increased levels of physical activity and decreased risks of breast and colorectal cancer. Overall, the paper is written in a very succinct manner and for an informed reader who already understands the field of MR. Given that this field is relatively new in epidemiology, this paper might be more readable if more interpretation of the study results were included in the discussion. Some specific examples are provided for consideration.

Specific Comments:

1. The association between physical activity and cancer incidence has been investigated in over 500 studies conducted worldwide. The authors have chosen to examine only breast and colorectal cancer in this analysis. Presumably, through the large consortia and the UK biobank data, they would have had access to data for other cancer sites as well. The authors should provide some rationale for why they have limited this investigation to just breast and colorectal cancers since there is now evidence for associations with at least 7-11 cancers. This rationale could be included in the introduction.
2. Results – the authors have provided results stratified by ER status but not by menopausal status or any other factor. In the methods section, later in this paper, it becomes evident that the authors did not have data available for any other effect modifiers that might be of interest including factors such as body mass index. Some explanation for why estrogen receptor status was the only effect modifier that was available to be considered for breast cancer could be added to the methods and results. Other important factors would be body mass index, family history of breast cancer as well as ethnicity/race.
3. Discussion – the literature that is cited in the discussion section on physical activity and cancer risk is somewhat selective and out-of-date and could be improved by including more updated reviews of the literature on physical activity and cancer that have been recently published. In addition, the description of the biologic mechanisms that could explain the association between physical activity and cancer risk cites only a few select papers and not always the most recent ones. Finally, the discussion is written very succinctly without much elaboration on the implications of these findings besides concluding that physical activity should be promoted for the primary prevention of these two cancer sites. The discussion would be more informative if a more fulsome assessment of the interpretation of these findings was provided.
4. Methods – the methods section could be enhanced with a more clear description of the source of the summary level data used for both exposure and cancer incidence.
5. References – an error in reference 28 was found for the author name which is likely the

Endogenous Hormone ... Group.

Reviewer #2:

Remarks to the Author:

In the literature there are studies suggesting the association between physical activity (PA) and the reduction of the risk of breast and colorectal cancers. The authors investigated their causal link by a Mendelian randomization approach.

The study is well conducted and presented. I suggested some minor comments.

1. Given the small number of GWA-significant SNPs used as instruments, they could try to investigate the presence of pleiotropy from a biological point of view.
2. They could provide some plots to investigate the directional pleiotropy.
3. I suggest to revise the notation, to be consistent.

Reviewer #3:

Remarks to the Author:

Studying whether the real association between physical activity and breast and colon cancers is causal or not, is very important to clinical practice. The authors of this study have access to great datasets to answer this question. However I cannot recommend this manuscript for publication in its current state. In particular the following flaws should be addressed:

=== 1. Choice of exposure instruments. ===

The authors should:

i) Use physical activity instruments from the 2018 Nature Comms paper "GWAS identifies 14 loci for device-measured physical activity and sleep duration". This would provide 5 SNPs with a p-value of $<5e-8$ (rather than just 2)

ii) just use SNPs of a P-Value of $5e-8$. If desperate to use less stringent SNPs, the authors should at least perform more detailed horizontal pleiotropy simulations to provide some reassurance on the choice of $1e-7$. The paper suggested just above also provides a template on performing horizontal pleiotropy simulations to justify lowering this threshold if need be (for a secondary analysis).

The following statement in the paper is very unconvincing -> "The intercept test from the MR-Egger regression was statistically significant in the analysis of colorectal cancer in women denoting potential pleiotropy; however, the corrected estimate from MR-Egger replicated the initial finding (Table 2)."

=== 2. For the main analysis, ensure the outcome GWAS does not include UK Biobank accelerometer participants ===

As indicated, the GECCO consortium includes 26,763 participants from the UK Biobank. The accelerometer participants from this consortium should be removed (meaning ~80% of the above participants can be included in the MR analysis). This should be the *main analysis*, not just a secondary/sensitivity analysis.

=== 3. Adjust for multiple testing. The authors are looking at multiple outcomes, and should adjust for this. ===

=== 4 Consider using exposure SNPs from a physical activity GWAS adjusted for BMI ===
The above Nature Comms paper has released physical activity SNPs that are adjusted for BMI too.

=== 5 Interpret results non-overoptimistically ===

i) The rectal cancer result is null, but the manuscript states:

"For colorectal subsite analyses, accelerometer-measured physical activity levels were inversely associated with risks of colon cancer (OR per 1 SD increment OR: 0.61, 95% CI: 0.47 to 0.79, P-value= 2×10^{-4}) and rectal cancer (OR: 0.76, 95% CI: 0.55 to 1.07, P-value=0.12)."

ii) There isn't an apparent statistically lower OR in men than women, but the manuscript states:

"The estimated effect size was stronger for women (OR: 0.54, 95% CI: 0.40 to 0.74, P-value= 1.2×10^{-4}) than men (OR: 0.82, 95% CI: 0.61 to 1.11, P-value=0.21)"

iii) ER-ve is null, yet the manuscript states:

"Similar magnitude inverse associations were found for estrogen receptor positive (ER+ve) (OR: 0.53, 95% CI: 0.35 to 0.82, P-value=0.004) and estrogen receptor negative (ER-ve) (OR: 0.78, 95% CI: 0.51 to 1.22, P-value=0.27) ..."

For instance iii) above it is interesting that using just the two genome wide significant SNPs ER-ve appears significant, whereas it ER-ve is null when using the 10 $p < 1e-7$ SNPs. Again it would be helpful to use a more powerful set of exposure SNPs here, as suggested above.

Finally, I would like to highlight a minor typo in the results text which refers to Figure 2 as being related to 'breast cancer' rather than 'colorectal cancer'.

International Agency for Research on Cancer

World Health
Organization

Reviewers' comments:

Reviewer #1 (Remarks to the Author): Expert in epidemiology

General Comments:

This large group of authors have written a novel and interesting paper on the association between physical activity and risks of breast and colorectal cancers using a Mendelian randomization (MR) analytic approach. This study was possible because they used the UK Biobank and GWA consortia data that included hundreds of thousands of study participants' physical activity and cancer incidence data. These types of large data sets have not been previously used to assess this particular research question. The major rationale for this analysis is that MR should overcome the confounding that exists in traditional analyses of epidemiologic studies that have relied on self-reported exposure data. Hence, there is interest in assessing if the associations observed in observational epidemiologic studies that have used self-reported data on physical activity and cancer incidence are also found in these MR analyses that should be free of these common biases. The authors conclude that there is support for a causal association between increased levels of physical activity and decreased risks of breast and colorectal cancer. Overall, the paper is written in a very succinct manner and for an informed reader who already understands the field of MR. Given that this field is relatively new in epidemiology, this paper might be more readable if more interpretation of the study results were included in the discussion. Some specific examples are provided for consideration.

Specific Comments:

1. The association between physical activity and cancer incidence has been investigated in over 500 studies conducted worldwide. The authors have chosen to examine only breast and colorectal cancer in this analysis. Presumably, through the large consortia and the UK biobank data, they would have had access to data for other cancer sites as well. The authors should provide some rationale for why they have limited this investigation to just breast and colorectal cancers since there is now evidence for associations with at least 7-11 cancers. This rationale could be included in the introduction.

We focused on two common cancers most strongly linked with self-reported physical activity from previous epidemiological studies and for which GWAS data were available on sufficiently large numbers of cases and controls. The World Cancer Research Fund/American Institute for Cancer Research (WCRF/AICR) classified the epidemiological evidence base linking physical activity and breast and colorectal cancers as 'strong', with only one other cancer (endometrial) similarly graded and the remaining cancers assessed as having weaker relationships with physical activity. We did not include endometrial cancer as there is ongoing work on such an analysis from another research group. We have now added text to the Introduction adding rationale for our decision to how physical activity relates to breast and colorectal cancers only.

P.10, paragraph 2; "Epidemiological studies have consistently observed inverse relationships between physical activity and risks of breast and colorectal cancer"²⁻⁵. The World Cancer Research Fund /American Institute for Cancer Research (WCRF/AICR) Continuous Update Project classified the evidence linking physical activity to lower risks of breast (post-menopausal) and colorectal cancer as 'strong'⁶."

2. Results – the authors have provided results stratified by ER status but not by menopausal status or any other factor. In the methods section, later in this paper, it becomes evident that the authors did not have data available for any other effect modifiers that might be of interest including factors such as body mass index. Some explanation for why estrogen receptor status was the only effect modifier that was available to be considered for breast cancer could be added to the methods and results. Other important factors would be body mass index, family history of breast cancer as well as ethnicity/race.

For our breast cancer analyses, we used publically available GWAS summary statistics from the Breast Cancer Association Consortium (BCAC) that are only available for overall breast cancer, and ER-positive and ER-negative breast cancer. These summary estimates are downloadable here:

<http://bcac.ccge.medschl.cam.ac.uk/bcacdata/oncoarray/gwas-icogs-and-oncoarray-summary-results/>.

Unfortunately, our use of summary data precluded analyses between physical activity and breast cancer risk by subgroups of other risk factors (e.g. BMI, exogenous hormone use, and menopausal status). We have outlined this limitation within the Discussion.

P.16; paragraph 2; “In addition, our use of summary-level data precluded subgroup analyses by other cancer risk factors (e.g. BMI, exogenous hormone use). We were also unable to stratify breast cancer analyses by menopausal status; however, the majority of women in the source GWAS had postmenopausal breast cancer¹².”

All participants included in the BCAC GWAS that our summary estimates were derived from were of European ancestry so analyses by ethnicity/race were not possible.

3. Discussion – the literature that is cited in the discussion section on physical activity and cancer risk is somewhat selective and out-of-date and could be improved by including more updated reviews of the literature on physical activity and cancer that have been recently published.

We have now updated the cited literature on previous epidemiological studies examining the associations between physical activity and breast and colorectal cancer. We now cite the participant-level pooled analysis of 12 prospective studies conducted by Moore et al. (JAMA Internal Medicine 2016 Jun 1;176(6):816-25) and the recent meta-analyses of all published prospective cohort data by the WCRF/AICR Continuous Update Project (<https://www.wcrf.org/dietandcancer>).

P.14, paragraph 2; “A large body of observational studies has investigated how physical activity relates to risk of breast and colorectal cancer¹⁵. In a participant-level pooled analysis of 12 prospective studies, when the 90th and 10th percentile of leisure-time physical activity were compared, lower risks of breast cancer (hazard ratio [HR]: 0.90, 95% CI: 0.87 to 0.93), colon cancer (HR: 0.84, 95% CI: 0.77 to 0.91), and rectal cancer (HR: 0.87, 95% CI: 0.80 to 0.95) were found³. Similarly, inverse associations between total physical activity and risks of postmenopausal breast and colorectal cancer were recently reported in meta-analyses of all published prospective cohort data by the WCRF/AICR Continuous Update Project^{16,17}.”

In addition, the description of the biologic mechanisms that could explain the association between physical activity and cancer risk cites only a few select papers and not always the most recent ones.

As suggested, we have now added references to two recent review articles that discussed the biological mechanisms that may underlie the physical activity and cancer association (Hojman et al., Cell Metabolism 2018 Jan 9;27(1):10-21 / Ulrich et al., Nat Rev Gastroenterol Hepatol 2018 Nov;15(11):683-698.

Finally, the discussion is written very succinctly without much elaboration on the implications of these findings besides concluding that physical activity should be promoted for the primary prevention of these two cancer sites. The discussion would be more informative if a more fulsome assessment of the interpretation of these findings was provided.

As suggested, we have now added an additional paragraph in the Discussion outlining the possible implications of our findings in terms of translation to daily activity levels.

P.15, paragraph 2; “There is currently no standard method in translating accelerometer data into energy expenditure values, such as metabolic equivalent of tasks (METs). However, it is estimated that a ~8 milli-gravity higher mean acceleration (roughly 1-SD in the current study) is approximately equivalent to a previously sedentary individual undertaking 15 minutes of vigorous activity (e.g. running) in a 24 hour period or 1 hour of moderate physical activity (e.g. fast walking) in a 24 hour period^{8,11,20}. In our study, we found that such an increase in daily activity translates to a 41% and 40% lower risks of developing breast and colorectal cancer, respectively.”

4. Methods – the methods section could be enhanced with a more clear description of the source of the summary level data used for both exposure and cancer incidence.

As suggested, we have now included additional text within the Methods section describing the GWAS used for the summary estimates for physical activity, breast cancer, and colorectal cancer.

P.17, paragraph 3; “Summary-level data were obtained from a recently published GWAS on accelerometer-measured physical activity conducted in 91,084 participants from the UK Biobank¹¹. In this GWAS, the regression models were adjusted for age, sex, the first ten genomic principal components, center, season (month), and genotyping chip. GWAS were performed with BOLT-LMM software, implementing a mixed-model linear regression that includes a random effect consisting of the SNPs other than the one being tested, and consequently taking into consideration relatedness among subjects¹¹. This GWAS identified 2 genome-wide-significant polymorphisms (P-value<5x10⁻⁸) associated with accelerometer-measured physical activity. The estimated SNP-based heritability was 14% suggesting that additional SNPs contributed to its variation. Consequently, for our primary analyses, we used a larger number of 10 independent (linkage disequilibrium [LD] r²≤0.001) genetic variants by relaxing the significance threshold to P-value<1x10⁻⁷.”

P.18, paragraph 2; “Summary data for the associations of the 10 accelerometer-measured genetic variants with breast cancer (overall and by estrogen receptor status: ER positive and ER negative) were obtained from a GWAS of 228,951 women (122,977 breast cancer [69,501 ER positive, 21,468 ER negative] cases and 105,974 controls) of European ancestry from the Breast Cancer Association Consortium (BCAC)¹². Details of the genotyping protocols in the BCAC are described elsewhere (iCOGS: <http://ccge.medschl.cam.ac.uk/research/consortia/icogs/>; OncoArray: <https://epi.grants.cancer.gov/oncoarray/>)^{12,42}. Genotyping data were imputed using the program IMPUTE214 with the 1000 Genomes Project Phase III integrated variant set as the reference panel. Single nucleotide polymorphisms (SNPs) with low imputation quality (imputation r²<0.5) were excluded. Top principal components (PCs) were included as covariates in regression analysis to address potential population substructure (iCOGS: top eight PCs; OncoArray: top 15 PCs) (Supplementary Table 1). For colorectal cancer, summary data from 98,715 participants (52,775 colorectal cancer cases and 45,940 controls) were drawn from a meta-analysis within the ColoRectal Transdisciplinary Study (CORECT), the Colon Cancer Family Registry (CCFR), and the Genetics and Epidemiology of Colorectal Cancer (GECCO) consortia^{13,43}. Imputation was performed using the Haplotype Reference Consortium (HRC) r1.0 reference panel and the regression models were further adjusted for age, sex, genotyping

platform (whenever appropriate), and genomic principal components (from 3 to 13, whenever appropriate) (Supplementary Tables 2, 3)."

5. *References – an error in reference 28 was found for the author name which is likely the Endogenous Hormone ... Group.*

We thank the reviewer for picking up this error. We have now updated this reference.

Reviewer #2 (Remarks to the Author): Expert in Mendelian randomisation

In the literature there are studies suggesting the association between physical activity (PA) and the reduction of the risk of breast and colorectal cancers. The authors investigated their causal link by a Mendelian randomization approach. The study is well conducted and presented. I suggested some minor comments.

We thank the reviewer for the positive feedback.

1. *Given the small number of GWA-significant SNPs used as instruments, they could try to investigate the presence of pleiotropy from a biological point of view.*

As well as statistical assessments for pleiotropy (MR-Egger, weighted median, MR-PRESSO) and visual inspection of scatter and funnel plots, we also examined how SNPs (and their proxies; $r^2 > 0.8$) included in the genetic instrument were associated with secondary phenotypes in Phenoscanner (<http://www.phenoscanter.medschl.cam.ac.uk/>) and GWAS catalog. We found that 5 of the 10 variants in the genetic instrument were also associated with adiposity-related traits. However, in sensitivity analyses our results were unchanged when we excluded the adiposity-related SNPs from the genetic instrument. This result therefore suggests that physical activity is associated with breast and colorectal cancer independently of adiposity.

2. *They could provide some plots to investigate the directional pleiotropy.*

As suggested, for visual assessment of possible directional pleiotropy, we now include scatter plots (showing the correlation of genetic associations of accelerometer-measured physical activity with genetic associations with cancer, and with regression slopes for the different models) and funnel plots in the Supplementary Materials (Supplementary Figures 2-5). These new figures, taken together with results from the statistical assessments for pleiotropy (MR-Egger, weighted median, MR-PRESSO), and our examination of how the genetic instruments were related to secondary phenotypes, provide little evidence that unbalanced pleiotropy has biased our causal estimates.

3. *I suggest to revise the notation, to be consistent.*

We have now checked and revised, where necessary, the notation for consistency.

Reviewer #3 (Remarks to the Author): Expert in GWAS

Studying whether the real association between physical activity and breast and colon cancers is causal or not, is very important to clinical practice. The authors of this study have access to great datasets to answer this question. However I cannot recommend this manuscript for publication in its current state. In particular the following flaws should be addressed:

=== 1. Choice of exposure instruments. ===

The authors should:

i) Use physical activity instruments from the 2018 Nature Comms paper "GWAS identifies 14 loci for device-measured physical activity and sleep duration". This would provide 5 SNPs with a p-value of $<5e-8$ (rather than just 2)

The 2018 Nature Comms GWAS indicated by the reviewer (Doherty et al., Nat Commun 2018 Dec 10;9(1):5257) was based on the same dataset (UK Biobank) and is of comparable size (~91,000 participants) to the GWAS we used for our MR analyses (Klimentidis et al., Int J Obes (London) 2018 Jun;42(6):1161-1176). The GWAS used in our analysis identified 2 genome-wide-significant polymorphisms ($P\text{-value} < 5 \times 10^{-8}$) associated with accelerometer-measured physical activity. The estimated SNP-based heritability was 14% suggesting that additional SNPs contributed to its variation. Consequently, for our primary analyses, we used a larger number of 10 independent (linkage disequilibrium [LD] $r^2 \leq 0.001$) genetic variants by relaxing the significance threshold to $P\text{-value} < 1 \times 10^{-7}$. The expanded number of genetic variants in the accelerometer-measured physical activity instrument allowed sensitivity analyses to be conducted to check for the influence of horizontal pleiotropy on the results.

In response to the reviewers suggestion, we have conducted sensitivity analyses using a new genetic instrument including the 5 SNPs identified in the GWAS ($P\text{-value} < 5 \times 10^{-8}$) by Doherty et al. Our results and conclusions are unchanged when we used this new instrument. As an example, for breast cancer an OR per 1 SD increment of 0.59 (95% CI: 0.42-0.84) was found in our primary analysis, with an OR of 0.51 (95% CI: 0.27-0.98) found for the instrument based on the GWAS by Doherty et al. For colorectal cancer, an OR per 1 SD increment of 0.60 (95% CI: 0.47-0.84) was found in our primary analysis, with an OR of 0.66 (95% CI: 0.48-0.90) found when using the instrument based on the GWAS by Doherty et al. These results are now presented in the Supplementary Materials (Supplementary Tables 13-17) and are described in the Methods and Results sections.

P.21, paragraph 2; "Finally, we also conducted a sensitivity analysis, conducting two-sample MR analyses with revised genetic instruments including variants (n=5) associated with accelerometer-measured physical activity (unadjusted and adjusted for BMI; $P\text{-value} < 5 \times 10^{-8}$) identified in another UK Biobank GWAS (n=91,105 participants)¹⁴. However, the MR results using the BMI adjusted GWAS estimates should be interpreted cautiously due to the potential for collider bias¹⁴."

P.13, paragraph 2; "Finally, a similar pattern of effect estimates for the accelerometer-measured physical activity and breast and colorectal cancer associations were found when a genetic instruments derived from another UK Biobank GWAS¹⁴ (unadjusted and adjusted for BMI) was used in the two-sample MR analyses (Supplementary Tables 13-17)."

ii) just use SNPs of a P-Value of $5e-8$. If desperate to use less stringent SNPs, the authors should at least perform more detailed horizontal pleiotropy simulations to provide some reassurance on the choice of $1e-7$. The paper suggested just above also provides a template on performing horizontal pleiotropy simulations to justify lowering this threshold if need be (for a secondary analysis).

The following statement in the paper is very unconvincing -> "The intercept test from the MR-Egger regression was statistically significant in the analysis of colorectal cancer in women denoting potential pleiotropy; however, the corrected estimate from MR-Egger replicated the initial finding (Table 2)."

As outlined to reviewer number 2, we now present scatter plots (showing the correlation of genetic associations of accelerometer-measured physical activity with genetic associations with cancer, and with regression slopes for the different models) and funnel plots (Supplementary Figures 2-5). These

new figures, taken together with results from the statistical assessments for pleiotropy (MR-Egger, weighted median, MR-PRESSO), and our examination of how the genetic instruments were related to secondary phenotypes, provide little evidence that unbalanced pleiotropy has biased our causal estimates.

Regarding the text from the previous version of the manuscript on the MR-Egger intercept test, in our updated colorectal cancer analyses (with the UK Biobank excluded from the colorectal cancer GWAS), no evidence of directional pleiotropy was found. We have now updated the manuscript accordingly.

P.13, paragraph 1; “The strength of the genetic instruments denoted by the F-statistic was ≥ 10 for all the accelerometer-measured physical activity variants and ranged between 29 and 56 (Table 1). Little evidence of directional pleiotropy was found for all models (MR-Egger intercept P-values > 0.06) (Table 2). The estimates from the weighted median approach were consistent with those of inverse variance weighted (IVW) models (Table 2).”

=== 2. For the main analysis, ensure the outcome GWAS does not include UK Biobank accelerometer participants ===

*As indicated, the GECCO consortium includes 26,763 participants from the UK Biobank. The accelerometer participants from this consortium should be removed (meaning ~80% of the above participants can be included in the MR analysis). This should be the *main analysis*, not just a secondary/sensitivity analysis.*

As suggested by the reviewer, we have now updated all of the analyses for colorectal cancer using GWAS summary estimates from the GECCO consortium with the UK Biobank excluded. Similar magnitude effect estimates were found to those from the original analyses.

=== 3. Adjust for multiple testing. The authors are looking at multiple outcomes, and should adjust for this. ===

As suggested, we have conducted false discovery rate (FDR; Q-value) correction using the Benjamini–Hochberg method. All of our inverse associations found between accelerometer-measured physical activity and breast and colorectal cancer remained statistically significant after this correction, with the exception of the result for rectal cancer (FDR Q-value of 0.05 from an initial P-value of 0.04). We have altered the text within the manuscript to describe our correction for multiple comparisons and to reflect the effect of this correction on the rectal cancer result.

P.19, paragraph 3; “FDR correction (Q-value) was performed using the Benjamini–Hochberg method⁴⁶.”

P.12, paragraph 2; “For colorectal subsite analyses, accelerometer-measured physical activity levels were inversely associated with risks of colon cancer (OR per 1 SD increment OR: 0.56, 95% CI: 0.42 to 0.73, FDR Q-value= 4.4×10^{-5}) and rectal cancer (OR: 0.68, 95% CI: 0.47 to 0.98, FDR Q-value=0.05), although the latter effect lost statistical significance after multiple comparison adjustment.”

=== 4 Consider using exposure SNPs from a physical activity GWAS adjusted for BMI ===

The above Nature Comms paper has released physical activity SNPs that are adjusted for BMI too.

In our main analyses, we chose not to use GWAS summary data for physical activity variants adjusted for BMI as using such effect estimates can introduce collider bias, and subsequent bidirectional effects, into downstream Mendelian randomization analyses. The risk of collider bias when using BMI adjusted estimates was also cited by Doherty et al. in the Nature Communications GWAS that the reviewer cites

(Doherty et al., Nat Commun 2018 Dec 10;9(1):5257), with the authors including a warning to interpret these adjusted estimates cautiously. However, as suggested by the reviewer, we now present a new supplementary table (Supplementary Table S17) with the MR results using the Doherty et al. 5 SNP instrument BMI adjusted estimates. Our causal effect estimates for physical activity and breast cancer and colorectal cancer remained largely unchanged. We have now added additional text in the Methods and Results section describing these results, and stated our belief that these results should be interpreted with caution due to the heightened risk of collider bias.

P.13, paragraph 2; “Finally, a similar pattern of effect estimates for the accelerometer-measured physical activity and breast and colorectal cancer associations were found when a genetic instruments derived from another UK Biobank GWAS¹⁴ (unadjusted and adjusted for BMI) was used in the two-sample MR analyses (Supplementary Tables 13-17).”

P.21, paragraph 2; “Finally, we also conducted a sensitivity analysis, conducting two-sample MR analyses with revised genetic instruments including variants (n=5) associated with accelerometer-measured physical activity (unadjusted and adjusted for BMI; P-value<5x10⁻⁸) identified in another UK Biobank GWAS (n=91,105 participants)¹⁴. However, the MR results using the BMI adjusted GWAS estimates should be interpreted cautiously due to the potential for collider bias¹⁴.”

=== 5 Interpret results non-overoptimistically ===

i) The rectal cancer result is null, but the manuscript states:

*"For colorectal subsite analyses, accelerometer-measured physical activity levels were inversely associated with risks of colon cancer (OR per 1 SD increment OR: 0.61, 95% CI: 0.47 to 0.79, P-value=2*10⁻⁴) and rectal cancer (OR: 0.76, 95% CI: 0.55 to 1.07, P-value=0.12)."*

For our updated analyses using colorectal cancer GWAS summary estimates with the UK Biobank excluded, inverse associations were found for both colon cancer (OR per 1 SD increment OR: 0.56, 95% CI: 0.42 to 0.73, FDR Q-value=4.4*10⁻⁵) and rectal cancer (OR: 0.68, 95% CI: 0.47 to 0.98, FDR Q-value=0.05). However, the rectal cancer association lost statistical significance after we corrected for multiple comparisons. We have now updated the text to reflect these updated results.

P.12, paragraph 2; “For colorectal subsite analyses, accelerometer-measured physical activity levels were inversely associated with risks of colon cancer (OR per 1 SD increment OR: 0.56, 95% CI: 0.42 to 0.73, FDR Q-value=4.4*10⁻⁵) and rectal cancer (OR: 0.68, 95% CI: 0.47 to 0.98, FDR Q-value=0.05), although the latter effect lost statistical significance after multiple comparison adjustment.”

ii) There isn't an apparent statistically lower OR in men than women, but the manuscript states:

*"The estimated effect size was stronger for women (OR: 0.54, 95% CI: 0.40 to 0.74, P-value=1.2*10⁻⁴) than men (OR: 0.82, 95% CI: 0.61 to 1.11, P-value=0.21)"*

Taking into account our updated colorectal cancer analyses we have now changed the text to describe the results by sex more clearly.

P.12, paragraph 2; “The inverse effect estimate was stronger for women (OR: 0.49, 95% CI: 0.35 to 0.68, FDR Q-value=1*10⁻⁴) than for men (OR: 0.76, 95% CI: 0.55 to 1.07, FDR Q-value=0.12), although this heterogeneity did not meet the threshold of significance (I²=66%; P-heterogeneity by sex=0.09).”

iii) ER-ve is null, yet the manuscript states:

"Similar magnitude inverse associations were found for estrogen receptor positive (ER+ve) (OR: 0.53, 95% CI: 0.35 to 0.82, P-value=0.004) and estrogen receptor negative (ER-ve) (OR: 0.78, 95% CI: 0.51 to 1.22, P-value=0.27) ..."

For instance iii) above it is interesting that using just the two genome wide significant SNPs ER-ve appears significant, whereas it ER-ve is null when using the 10 $p < 1e-7$ SNPs. Again it would be helpful to use a more powerful set of exposure SNPs here, as suggested above.

We have now updated the text to better reflect the results by breast cancer subtype.

P.11, paragraph 3; "An inverse association reaching the threshold of statistical significance was only found for estrogen receptor positive breast cancer (ER^{+ve}) (OR: 0.53, 95% CI: 0.35 to 0.82, FDR Q-value=0.007), and not estrogen receptor negative (ER^{-ve}) breast cancer (OR: 0.78, 95% CI: 0.51 to 1.22, FDR Q-value=0.27); although this heterogeneity by subtype was not statistically different ($I^2=25\%$; P-heterogeneity by subtype=0.25)."

As indicated above, we have provided an additional sensitivity analysis using the genetic variants identified in the GWAS by Doherty et al. The effect estimates yielded using this new instrument were of similar direction and magnitude to our primary analyses.

Finally, I would like to highlight a minor typo in the results text which refers to Figure 2 as being related to 'breast cancer' rather than 'colorectal cancer'.

We thank the reviewer for picking up this error. We have now changed the results text to refer correctly to each figure.

Reviewers' Comments:

Reviewer #1:

Remarks to the Author:

The authors have adequately addressed the questions and comments that I raised in the initial review. I have no further suggested edits and thank them for the comprehensive response to my comments.

Reviewer #2:

Remarks to the Author:

The authors addressed very well to my issues. No further issues arise.

Reviewer #3:

Remarks to the Author:

I would like to thank the authors for a thorough response to all the reviewer comments. However, there are still three issues to address:

1) I still don't agree with the rationale for selecting exposure instruments using a $p < 1 \times 10^{-7}$ threshold.

The authors should select SNPs at the more widely accepted $p < 5 \times 10^{-8}$ threshold. In doing this, they should select the most SNPs available (i.e. use 5 SNPs, not 2 SNPs).

It's fine to use the other 10 SNPs (at $p < 1 \times 10^{-7}$) in a sensitivity analysis. Although one could argue why not select ~50 SNPs at $p < 1 \times 10^{-6}$, or use even more instruments at an even lower threshold... This is why it's really important that the main analysis should be reported using the 5 genome wide significant SNPs.

2) Null results should be more clearly stated in the text

For example, it is a little evasive to say that "physical activity levels were inversely associated with risks of ... rectal cancer ... although the latter effect lost statistical significance after multiple comparison adjustment"

Instead this should be reworded to say "There was weak evidence for an inverse association between physical activity levels and rectal cancer (OR...)".

Please use similar wording to the above for the women vs men results (p.12, paragraph 2).

3) Update discussion paragraph outlining the possible implications of findings in terms of translation to daily activity levels

I'm not sure references 8, 11, and 20 actually give guidance on a 1SD increase equating to 15 mins extra vigorous activity or 1 hour moderate activity. Given the public health importance of accurately reporting these numbers, I therefore recommend the authors update the quoted figures to reflect the below:

I've had a look at the UK Biobank showcase, and using an accepted 100mg threshold for moderate activity (see <http://biobank.ndph.ox.ac.uk/showcase/field.cgi?id=90127>), a 1 SD increase in equates to approximately 50 minutes extra moderate activity per week.

Similarly a 425mg threshold for vigorous activity indicates a 1 SD increase equating to approximately just 8 minutes extra vigorous activity per week.

International Agency for Research on Cancer

World Health
Organization

Reviewers' comments:

Reviewer #1 (Remarks to the Author):

The authors have adequately addressed the questions and comments that I raised in the initial review. I have no further suggested edits and thank them for the comprehensive response to my comments.

We thank the reviewer for the positive feedback on our previous response.

Reviewer #2 (Remarks to the Author):

The authors addressed very well to my issues. No further issues arise.

We thank the reviewer for the positive feedback on our previous response.

Reviewer #3 (Remarks to the Author):

I would like to thank the authors for a thorough response to all the reviewer comments. However, there are still three issues to address:

1) I still don't agree with the rationale for selecting exposure instruments using a $p < 1 \times 10^{-7}$ threshold.

The authors should select SNPs at the more widely accepted $p < 5 \times 10^{-8}$ threshold. In doing this, they should select the most SNPs available (i.e. use 5 SNPs, not 2 SNPs).

It's fine to use the other 10 SNPs (at $p < 1 \times 10^{-7}$) in a sensitivity analysis. Although one could argue why not select ~50 SNPs at $p < 1 \times 10^{-6}$, or use even more instruments at an even lower threshold... This is why it's really important that the main analysis should be reported using the 5 genome wide significant SNPs.

For our study, we followed an *a priori* analysis plan in which for our main analysis we used genetic variants and estimates associated with accelerometer-measured physical activity identified in the UK Biobank GWAS conducted by Klimentidis et al (Klimentidis et al., *Int J Obes [London]* 2018 Jun;42[6]:1161-1176). This GWAS identified two variants associated with accelerometer-measured physical activity ($r^2 = 0.1\%$) at the genome-wide significance threshold ($P\text{-value} < 5 \times 10^{-8}$). The estimated SNP-based heritability for accelerometer-measured physical activity in this GWAS was 14%, suggesting that additional SNPs contributed to its variation. Consequently, for our main analyses, we used a larger number of ten independent (linkage disequilibrium [LD] $r^2 \leq 0.001$) genetic variants by relaxing the significance threshold to $P\text{-value} < 1 \times 10^{-7}$. The expanded number of genetic variants gave us a stronger accelerometer-measured physical activity instrument ($r^2 = 0.4\%$), and allowed us to conduct sensitivity analyses to test for the influence of horizontal pleiotropy (MR-Egger, weighted median, MR-PRESSO). As an additional sensitivity analysis, we also conducted analyses using an instrument comprised of the two genome-wide significant variants identified in this GWAS, and we present these results together with our main results in Table 2. Importantly, our results were similar for both the ten and two variant instruments.

Our analytical approach, of gaining a stronger genetic instrument by relaxing the GWAS significance threshold for variant inclusion, is consistent with that taken in other MR analyses that have examined the health effects of accelerometer-measured physical activity on depression (Choi et al., *JAMA Psychiatry* 2019 Apr 1;76(4):399-408) and various other major diseases and risk factors (Doherty et al., *Nat Commun* 2018 Dec 10;9(1):5257). The reviewer suggests that for our main analyses we use an

instrument comprised of the five genetic variants identified at $P\text{-value} < 5 \times 10^{-8}$ in the Doherty et al GWAS for accelerometer-measured physical activity (Doherty et al., Nat Commun 2018 Dec 10;9(1):5257). This GWAS was based on the same dataset (UK Biobank) and was of comparable size (~91,000 participants) to the GWAS we used for our main MR analyses (Klimentidis et al., Int J Obes [London] 2018 Jun;42[6]:1161-1176). Furthermore, it is noteworthy that for the MR analyses for various major diseases and risk factors conducted by Doherty et al in their GWAS paper, similar to our approach, they relaxed the P-value inclusion threshold for the accelerometer-measured physical activity instrument (their instrument included >60 variants associated at $P\text{-value} < 5 \times 10^{-6}$). Like us, the rationale for relaxing the P-value threshold was that the revised instrument explained more phenotypic variance with negligible effects on horizontal pleiotropy.

In light of the reviewers last set of comments we conducted additional sensitivity analyses using an instrument comprised of the five genome-wide significant SNPs identified in the GWAS by Doherty et al. Importantly, our results and conclusions are unchanged when results from these sensitivity analyses are taken into consideration. As an example, for breast cancer an OR per 1 SD increment of 0.59 (95% CI: 0.42-0.84) was found in our main analysis, with an OR of 0.51 (95% CI: 0.27-0.98) found for the instrument based on the GWAS by Doherty et al. For colorectal cancer, an OR per 1 SD increment of 0.60 (95% CI: 0.47-0.84) was found in our main analysis, with an OR of 0.66 (95% CI: 0.48-0.90) found when using the instrument based on the GWAS by Doherty et al. These results are presented in the Supplementary Materials (Supplementary Tables 13-17) and are described in the Methods and Results sections.

Overall, it is our belief that the main results in our paper should remain those derived using the ten variant instrument which explains more of the variance in accelerometer-measured physical activity than the genome-wide significant variants, and allowed us to conduct robust sensitivity analyses to assess the possible influence of horizontal pleiotropy. The consistent results we found for these and other sensitivity analyses contained within the main paper and supplementary materials reassure us on the robustness of the inverse associations we found between accelerometer-measured physical activity and risks of breast and colorectal cancer.

2) Null results should be more clearly stated in the text

For example, it is a little evasive to say that "physical activity levels were inversely associated with risks of ... rectal cancer ... although the latter effect lost statistical significance after multiple comparison adjustment"

Instead this should be reworded to say "There was weak evidence for an inverse association between physical activity levels and rectal cancer (OR...)"

As suggested, we have now altered the text to better reflect the rectal cancer association.

P.13, paragraph 2; "For colorectal subsite analyses, accelerometer-measured physical activity levels were inversely associated with risks of colon cancer (OR per 1 SD increment OR: 0.56, 95% CI: 0.42 to 0.73, FDR Q-value= 4.4×10^{-5}); while there was weak evidence for an inverse association between accelerometer-measured physical activity levels and rectal cancer (OR: 0.68, 95% CI: 0.47 to 0.98, FDR Q-value=0.05)."

Please use similar wording to the above for the women vs men results (p.12, paragraph 2).

As suggested, we have now altered the text to better reflect the women versus men results.

P.13, paragraph 2; "For colorectal cancer, a 1 SD increment in accelerometer-measured physical

activity level was associated with a 40% lower risk (OR: 0.60, 95% CI: 0.47 to 0.76, FDR Q-value=1*10⁻⁴). The inverse effect estimate was stronger for women (OR: 0.49, 95% CI: 0.35 to 0.68, FDR Q-value=1*10⁻⁴), while there was weak evidence for an inverse association for men (OR: 0.76, 95% CI: 0.55 to 1.07, FDR Q-value=0.12); this heterogeneity did not meet the threshold of significance (I²=66%; P-heterogeneity by sex=0.09)."

3) Update discussion paragraph outlining the possible implications of findings in terms of translation to daily activity levels

I'm not sure references 8, 11, and 20 actually give guidance on a 1SD increase equating to 15 mins extra vigorous activity or 1 hour moderate activity. Given the public health importance of accurately reporting these numbers, I therefore recommend the authors update the quoted figures to reflect the below:

I've had a look at the UK Biobank showcase, and using an accepted 100mg threshold for moderate activity (see <http://biobank.ndph.ox.ac.uk/showcase/field.cgi?id=90127>), a 1 SD increase in equates to approximately 50 minutes extra moderate activity per week.

Similarly a 425mg threshold for vigorous activity indicates a 1 SD increase equating to approximately just 8 minutes extra vigorous activity per week.

We thank the reviewer for his comments on our text attempting to translate our results to daily activity levels. As we state in the text there is currently no standard method in translating accelerometer data into energy expenditure values, such as metabolic equivalent of tasks (METs). The values we cited were based on calculations conducted by Choi et al in their physical activity and depression MR paper (eTable 13 in the supplementary materials; Choi et al., JAMA Psychiatry 2019 Apr 1;76(4):399-408). That said, in response to the reviewer comment, we are happy to revise these estimates in line with those presented in the UK Biobank showcase and have updated the text accordingly.

P.16, paragraph 2; "There is currently no standard method in translating accelerometer data into energy expenditure values, such as metabolic equivalent of tasks (METs). However, using an accepted threshold for moderate activity (e.g. fast walking) of 100 milli-gravity^{19,20}, 1-SD higher mean acceleration (~8 milli-gravity) equates to approximately 50 minutes extra moderate activity per week. Similarly, using an accepted threshold of 425 milli-gravity for vigorous activity (e.g. running)^{19,20}, a 1-SD higher mean acceleration equates to approximately 8 minutes of extra vigorous activity per week. In our study, we found that such an increase in weekly activity translates to a 41% and 40% lower risks of developing breast and colorectal cancer, respectively."

Reviewers' Comments:

Reviewer #3:

Remarks to the Author:

The authors have addressed two of my three comments. I am very keen to see this manuscript published, as it will be well cited and influential in the field. As a result, the methodology used in this paper will influence many future efforts in this space. It is therefore critical that the analysis is conducted in the strongest manner possible, rather than merely following what other manuscripts have done in the past.

Stephen Burgess, George Davey Smith and colleagues recently published a Wellcome Open Research article on 'guidelines for performing Mendelian randomization investigations'. The article recognises that the most important decision to be made in designing a Mendelian randomization investigation is which genetic variants to include in the analysis.

In the case of physical activity we ideally want to select all variants that are significantly associated with the exposure of interest i.e. at $p < 5 \times 10^{-8}$. Sensitivity analysis should then be conducted at less stringent criteria, where explained variation is greater but also the chances of horizontal pleiotropy are also increased.

This would require minimal changes to the manuscript, and importantly ensure the reported results are methodologically sound (i.e. less prone to horizontal pleiotropy).

Therefore, I strongly feel the presentation of current manuscript should be changed as follows:

1) For the main analysis and reporting, the authors use the most available genome wide significant instruments at a p-value threshold of $p < 5 \times 10^{-8}$

2) For the sensitivity analysis and reporting, the authors can select instruments with a p-value threshold of $p < 1 \times 10^{-7}$

Reviewers' comments:

Reviewer #3 (Remarks to the Author):

The authors have addressed two of my three comments. I am very keen to see this manuscript published, as it will be well cited and influential in the field. As a result, the methodology used in this paper will influence many future efforts in this space. It is therefore critical that the analysis is conducted in the strongest manner possible, rather than merely following what other manuscripts have done in the past.

Stephen Burgess, George Davey Smith and colleagues recently published a Wellcome Open Research article on 'guidelines for performing Mendelian randomization investigations'. The article recognises that the most important decision to be made in designing a Mendelian randomization investigation is which genetic variants to include in the analysis.

In the case of physical activity we ideally want to select all variants that are significantly associated with the exposure of interest i.e. at $p < 5 \times 10^{-8}$. Sensitivity analysis should then be conducted at less stringent criteria, where explained variation is greater but also the chances of horizontal pleiotropy are also increased.

This would require minimal changes to the manuscript, and importantly ensure the reported results are methodologically sound (i.e. less prone to horizontal pleiotropy).

Therefore, I strongly feel the presentation of current manuscript should be changed as follows:

1) For the main analysis and reporting, the authors use the most available genome wide significant instruments at a p-value threshold of $p < 5 \times 10^{-8}$

2) For the sensitivity analysis and reporting, the authors can select instruments with a p-value threshold of $p < 1 \times 10^{-7}$

As suggested by the reviewer, we now present the analyses using the 5 SNP genome-wide significant instrument (identified by Doherty et al., Nat Commun 2018 Dec 10;9[1]:5257) as the main results, and the analyses using the extended 10 SNP instrument (identified by Klimentidis et al., Int J Obes [London] 2018 Jun;42[6]:1161-1176) as secondary analyses. Results for both of these instruments are now presented in an updated Table 2.